# Validation of Novel Stride Length Model-Based Approaches to Estimate Distance Covered Based on Acceleration and Pressure Data During Walking

**DOI:** 10.3390/s25196217

**Published:** 2025-10-07

**Authors:** Armelle-Myriane Ngueleu, Martin J.-D. Otis, Charles Sebiyo Batcho

**Affiliations:** 1Center for Interdisciplinary Research in Rehabilitation and Social Integration, Centre Intégré Universitaire de Santé et de Services Sociaux de la Capitale-Nationale, School of Rehabilitation Sciences, Faculty of Medicine, Université Laval, 525 Blvd. Wilfrid-Hamel, Quebec City, QC G1M 2S8, Canada; charles.batcho@fmed.ulaval.ca; 2Automation and interactive Robotic Laboratory (AIRL), Department of Applied Science, Université de Quebec à Chicoutimi, 555 Blvd. of University, Chicoutimi, QC G7H 2B1, Canada; martin_otis@uqac.ca

**Keywords:** 6-MWT, novel approaches, smart insoles, stride length, total distance

## Abstract

The ability to walk is essential in daily life, making walking outcomes key measures in clinical practice. This study aims to develop and validate two novel stride length model-based approaches for total distance estimation using a smart insole. Eight participants wore a pair of smart insoles. For a period of six minutes, each participant walked back and forth on a predefined 20 m pathway, and the numbers of round trips and strides taken were counted. Two stride length estimation approaches based on the director coefficients of acceleration data (Approach 1) and dynamic time warping (Approach 2) using smart insoles were used. The median accuracies of the total distance using Approach 1 are 98.92% [1.24%] (ICC = 0.992) and 98.69% [2.44%] (ICC = 0.994) for the right and left sides, respectively. For Approach 2, the average accuracies are 98.95% [0.18%] (ICC = 0.996) for the right side and 99.03% [0.14%] (ICC = 0.991) for the left side. The Mann–Whitney U test shows no statistically significant difference between the actual distance and smart insole for the total distance covered. Furthermore, there is no statistically significant difference between Approach 1 and Approach 2 for stride length. Although the sample size was small, the estimated total distance using the novel model-based algorithms appears to be accurate in comparison to the actual total distance.

## 1. Introduction

Strong evidence showed that physical activity, including walking, has substantial health benefits [1,2,3,4]. Walking is the most commonly reported activity among adults who meet the recommended levels of physical activity [5]. Walking will become increasingly important for public health as the population ages [6]. It is also relevant in the fight against obesity and in increasing daily caloric expenditure. Several parameters can be measured during walking, such as number of strides/steps, distance covered, walking speed, etc.

Different measurement techniques and tools are used in clinical settings and in the community to assess people’s walking ability. For example, in the community, the distance covered is a parameter that is frequently measured for athletes. In a clinical setting, the primary goal of rehabilitation after a stroke is to restore the patient’s ability to walk. Gait analysis can be used at various stages to achieve the following: for the initial assessment of the degree of impairment and to guide the clinician in prescribing treatment; for monitoring rehabilitation outcomes; and for the final assessment of the benefits of treatment [7]. Assessing the distance covered by a patient enables healthcare professionals to better assess the effectiveness of the provided intervention and the patient’s physical recovery progress. Such assessment can be performed through clinical tests, particularly the six-minute walk test (6-MWT) [8].

Distance and stride length estimation can be performed using different spatio–temporal analysis sensors, the most accurate of which are motion capture systems [9,10]. Wearable instrumented technologies based on a single sensor, such as accelerometers, are often used in outdoor settings to estimate the stride length and distance covered. However, the accuracy of these results is limited [11,12]. In fact, studies have shown that accelerometers alone do not provide accurate results for total distance and stride length [11,12]. The combination of two or more additional sensors, such as an accelerometer and a gyroscope, can improve the accuracy of distance and stride length. The PODOSmart insole, equipped with an inertial sensor, estimates stride length based on artificial intelligence algorithms [13]. The Global Positioning System (GPS) is another technology commonly used to determine the distance covered, although distances provided by some GPS devices are underestimated compared to the actual distance covered. Particularly, in indoor settings, several conditions affect the accuracy of the distance estimation, which varies over time [14,15,16].

There are other different approaches to estimate the total distance covered [17,18,19,20,21,22,23]. In clinical setting, the total distance covered during the six-minute walking test can be obtained by multiplying the average stride length by the number of strides. In the literature, stride length is estimated by various methods [24,25,26], which can be divided into two types: direct and indirect methods. The direct method uses the double integration of acceleration [20,23]. The indirect methods use either gait models based on biomechanical models [27,28], empirical relationship models [29,30,31], and inverted pendulum models [32,33,34], or statistical regression models using the relationship between stride length and sensor data [35,36,37,38]. For example, the stride length was estimated based on a linear combination of walking frequency and acceleration variance [21], a linear regression model of stride frequency and stride velocity [22], and an extended/adaptive Kalman filter [18,19]. Another approach for estimating stride length is based on foot displacement [39]. In that study [39], Suzuki et al. estimated the foot trajectories using a quaternion to represent the IMU orientation, and the stride length was calculated from the foot position absolute value.

Although the direct method is one of the best methods in theory, it contains noise, bias, and drift in acceleration data despite the zero-velocity assumption for a reasonable initialization in the integration of acceleration data for each stance phase [37]. In Peruzzi et al. (2011) [40] and Skog et al. (2010) [41], the authors used double integration of foot acceleration with drift compensation at particular points where no velocity was used for stride length estimation. In other studies, drift was compensated for, using a linear model [11,12,42]. An improved stride length estimation approach was developed based on the zero-velocity assumption, acceleration, and angular velocity dedrifting combination [9,10,43,44]. Mariani et al. [9] used a nonlinear drift model, more accurately for the stride length. Rampp et al. [43] used an individualized drift model for stride length estimation. Zrenner et al. [10] successfully estimated stride length with 7.9% of error based on the determination of a continuous integration value correlated with the foot velocity, a stride segmentation using an initial ground contact and a regression model to translate this integration value into a velocity value. The calculation of the foot velocity used in Zrenner et al. [10] was based on a polynomial function of second order, where the constants were estimated on a set of training data with known reference velocity observations using parametric regression analysis [45]. The stride length error reported in [10] can be reduced.

Parametric and nonparametric models exist for the indirect method, using statistical regression models. The most common approach for parametric models is linear regression between the stride length and walking characteristic features. Although nonparametric models are more accurate than parametric models, they require a large number and wide variety of labelled data using a deep learning approach for stride length estimation [46,47,48,49,50,51].

The purpose of the present study is to develop and validate novel model-based algorithms, adapted to smart insoles, to measure stride length and thus estimate total distance walked, with acceleration and pressure data. The assumption is that the approach based on dynamic time warping (DTW) would be more accurate than the approach based on the director coefficients of acceleration data to estimate the total distance, based on the average length stride and number of strides, for clinical requirements. In fact, DTW is an algorithm that measures the similarity between two temporal sequences by finding an optimal alignment that minimizes the cumulative distance between corresponding points regardless of speed and duration, whereas the acceleration director coefficient is a mathematical model.

## 2. Materials and Methods

### 2.1. Data Collection

Data from 8 healthy participants (6 males and 2 females), aged 39.8 ± 17.56 years old (range: 21 to 68), weight of 74.63 ± 11.40 kg, height of 173 ± 10.42 cm, and foot size of 29.61 ± 2.25 cm (ranging from 29 to 32 cm) are used in this study. Each participant wore a pair of smart insoles, I-SOL (Gilon, Seongnam, Republic of Korea), and two Gait Up monitors (Gait Up; Lausanne, Switzerland). For synchronization, the smart insole and Gait Up monitors were reset at the same time. Then, they performed the six-minute walking test (6-MWT) on the predefined distance of 20 m pathway back and forth (see Figure 1). The number of round trips were calculated directly for a single foot, and the number of strides (Ns) were determined by a manual step counter using video recording as gold standard. Each participant indicated in advance which foot they would use to start walking, and fluorescent Velcro was attached to the shin of that foot. During the walk, team members incremented the manual counter each time the participant’s Velcro-equipped foot took a stride. For each participant, the total distance was calculated by multiplying the number of round trips by 40 m for a single foot and was used as the actual distance. Once 6-MWT had elapsed, the participant stopped, and a team member measured the distance between the cone and the participant using an odometer. Similarly, if a half turn had also been completed, this was also added. These distances were added to the number of round trips multiplied by 40 m to determine the total distance covered. The number of strides taken was reported in a previous study by Ngueleu et al. [52]. The estimated total distance (*Td*) using the smart insoles is determined as in Equation (1). The average stride length is estimated using the model-based algorithms described in the following section. The accuracy of the two approaches and Gait Up monitors is determined as in Equation (2), with the outcome representing the total distance (*Td*) or average stride length (*L*).*Td = LNs*,(1)*Accuracy* = (1 − (|*actual outcome* − *estimated outcome*|/*actual outcome*)) × 100%(2)

### 2.2. Foot-Worn Sensors

The smart insole ISOL (Gilon, Seongnam, Korean), integrating a 3D accelerometer and four pressure sensors are located mainly under the heel, the great toe, the first, and the fifth metatarsal bones (see Figure 2). The smart insole is a small, low-power, stand-alone device that integrates a microcontroller, Bluetooth communication, and battery. The electronic module is in the midfoot. The smart insoles transmit data via Bluetooth to an application embedded in a tablet or mobile phone. Four US foot sizes, from 8” to 12” of the smart insoles, were used in this study. The acceleration and pressure data are sampled at 40 Hz.

Two Gait Up monitors (firmware version 4.2.5.3, Physilog, Lausanne, Switzerland) were attached to both shoes (see Figure 3). The Gait Up monitors are composed of inertial measurement units embedded with an accelerometer, a gyroscope, and a barometer. The sampling rate of the Gait Up monitors is 200 Hz. The Gait Up monitors were validated for stride length using an eight camera motion capture system (Vicon, Oxford Metrics, Oxford, UK, 200 Hz), which was considered as reference system by Schwameder et al. [53].

### 2.3. Phase Segmentation of Signal

When it comes to smart insoles, gait events around the heel-strike and toe-off were automatically identified using the antero–posterior acceleration, the pressure data under the heel and great toe (see Figure 4 and Figure 5). A toe-off event appeared just at the end of the maximum pressure under the toe, and a heel-strike event occurred at the start of the heel pressure increase [54]. On the acceleration data, the toe-off event fits with the antero-posterior acceleration increasing start and the heel-strike event matches with the maximum antero–posterior acceleration peak [55]. For identification of toe-off events, the function “findpeaks” in the MATLAB software version R2021b (MathWorks, Natick, MA, USA) is used on the pressure under the great toe. The function “islocalmax” is used on the antero–posterior acceleration data for the identification of heel-strike. In fact, the function “islocalmax” returns a logical array, whose elements are 1 (true) when a local maximum is detected in the corresponding element of the acceleration. The function “find” is then used to return a vector containing the linear indices of each non-zero element in logical array. The function “vertcat” is finally used to concatenate arrays vertically.

When it comes to the polynomial model, the trajectories of the 3D positions were obtained using the OptiTrack motion capture system (version 1.1, NaturalPoint, Corvallis, OR, USA). Six markers were positioned on the metatarsals (n = 3), cuboid, calcaneus, and talus of a participant. A participant performed six strides at a predefined distance of 3 m back and forth pathway. The sampling rate is 100 Hz. In accordance with the literature, toe-off and heel-strike events were identified in 3D position data [56,57,58,59]. Each toe-off event was defined as the instant of minimum resulting vertical displacement in the time range of 0.1 s to 0.8 s (as seen in Figure 6), and used to segment consecutive gait cycles on vertical position trajectory. Each heel-strike was defined as the instant of the ceiling from the antero–posterior displacement (see Figure 6) [60].

### 2.4. Stride Length Model-Based Algorithms

Approach 1: This approach is based on the director coefficients of acceleration data using a reference model and smart insoles signals (acceleration and force sensors). The data of the reference model and smart insoles are measured on three axes (3D). However, only the antero–posterior direction is used in this approach. The displacement of position (*Pm*(*t*)) for the reference model (subscripted m to describe the model) can be represented as a 5th order polynomial. In this study, the initial position is zero for each toe-off, and the initial and final velocities are also zero [61], allowing to find the final position for the heel-strike. Therefore, the displacement of position becomes a 3rd order polynomial, described in Equation (3), defined as a vector of three elements:*P*_3 × *nm*_(*t*) = *c*_1_*t*^3^ + *c*_2_*t*^2^ + *c*_3_*t* + *c*_4_ = [*P_mx_ P_my_ P_mz_*]*^T^*(3)
where cj is a vector of constants (with 1 ≤ j ≤ 4) to find, t is the time over a window (segmented signal of the model) of n samples, T is the period of the corresponding stride cycle defined by two adjacent toe-off events, and *Pm*(*t*) is the resulting position model. The first and second derivatives make it possible to find the velocity *Vm*(*t*), described in Equation (4) and the acceleration *Am*(*t*) described in Equation (5), respectively. The representation of *Vm*(*t*) as a second order polynomial was reported in some studies [10,45].*V*_3 ×_ *_nm_* (*t*) = 3*c*_1_*t*^2^ + 2*c*_2_*t* + *c*_3_ = [*V_mx_ V_my_ V_mz_*]*^T^*(4)*A*_3 × *nm*_ (*t*) = 6*c*_1_*t* + 2*c*_2_ = [*A_mx_ A_my_ A_mz_*]*^T^*(5)

From these Equations (3)–(5), only *x*-axis (antero–posterior axis) is used *Pmx*, *Vmx*, and *Amx*. For the Approach 1, the ratio is calculated with the director coefficients of the antero–posterior acceleration of the smart insole (*Asx*(*t*), using subscript s for describing the signal acquired from the smart insole) and the reference model (*Amx*) illustrated in Eqaution (5). The director coefficient of the reference model is 6c_1_. For the smart insole, the straight line that best fits with the swing phase data (*Asxn*(*t*)) for n samples was determined based on the antero–posterior acceleration (*Asx*(*t*)) for each gait cycle. The “lsqcurvefit” function was used to determine the director coefficient of *Asxn*(*t*)) for each gait cycle. The director coefficients have no units. This ratio (*R*) is then used to estimate the average stride length (*L*), as illustrated in Algorithm 1.
**Algorithm 1**: Model-based algorithm using director coefficients c_j_ of acceleration data for stride length**Input:** Acceleration *A*_3_
_×_
_ns_(*t*) *=* [*A_sx_*, *A_sy_*, *A_sz_*]*^T^*, *N_s_* stride count for the insole, and position *P*_3_
_×_
_nm_ (*t*) *=* [*P_mx_*, *P_my_*, *P_mz_*]*^T^* for the reference model**Output**: Averaged stride length (*L*) 1: Determine the coefficients of the model position *P_mx_*(*t*) in Equation (3)2: Calculate velocity *V_mx_* (*t*) in Equation (4)3: Calculate acceleration *A**_m_**_x_* (*t*) of model in Equation (5)4: **For** number of strides (N_s_) **do**  5: Segmentation, centering, and determination of the straight line of *A_sxn_*(*t*) of insole  6: Determine the ratio of director coefficients (*R*) of the accelerations of model and insole  7: Calculate mean of this ratio (*R*)8: **End for**9: Return *R*10: Calculate average stride length (*L*) by multiplying *P_x_* and *R*

Approach 2: This approach is based on the dynamic time warping (DTW) algorithm, which aligns the acceleration data of the reference model with that of the smart insoles. In general, the DTW is an algorithm which can be used to estimate the distance between two data sequences by aligning points [62]. In this approach, DTW is used to estimate the distance by aligning the antero–posterior accelerations of the reference model and the smart insoles. In addition, the coefficient (*S*) for n samples in Equation (6) is calculated as follows:*S* = 1/||*A_sxn_*||(6)
where ||*A_sxn_*|| represents the absolute value of the antero–posterior acceleration during the swing phase. The estimation of the DTW value (*d*(*t*)) between the acceleration data of the reference model (*Amxn*(*t*)) and the smart insole (*Asxn*(*t*)) for n samples is illustrated in Equation (7) below:*d_n_*(*t*) = *DTW*(*A_mxn_*(*t*),*A_sxn_*(*t*))(7)
Next, Equation (8) calculated the coefficient (*S*) multiplied by mean of *d*(*t*) as illustrated below:*E* = *mean*(*d_n_*(*t*))*S*(8)
where *A_mxn_* represents the antero–posterior acceleration of the reference model for n samples and *A_sxn_* represents the antero–posterior acceleration of the smart insole for n samples. The average stride length (L) is calculated by multiplying E by the position Px of the model as illustrated in Algorithm 2.
**Algorithm 2**: Model-based algorithm using the DTW for stride length**Input:** Acceleration *A*_3_
_×_
_nm_ (*t*) = [*A_mx_*, *A_my_*, *A_mz_*]*^T^*, *N_s_* stride count for the insole, and position *P*_3_
_×_
*_nm_* (*t*) = [*P_mx_*, *P_my_*, *P_mz_*]*^T^* for the reference model**Output**: Averaged stride length (*L*), 1: Determine the coefficients of the model position *P**_m_**_x_*(*t*) in Equation (3)2: Calculate velocity *V**_m_**_x_*(*t*) of model in Equation (4)3: Calculate acceleration *A**_m_**_x_*(*t*) of model in Equation (5)4: Determine coefficient (*S*) in Equation (6)5: **For** number of strides **do**  6: Segmentation, centering, and determination of the straight line of *A_sxn_*(*t*)  7: Calculate dn(t) in Equation (7)  8: Calculate E in Equation (8)9: End for10: Return *E*11: Determine *L* by multiplying *P*_x_ and *E*

### 2.5. Statistical Analysis

The average stride length and total distance covered by each participant were calculated separately for the right and left sides for each approach. Thus, the total distance was estimated as illustrated in (1). The intraclass correlation coefficient (ICC) values and confidence intervals (CI) were calculated to determine the relative validity of each stride length and the total distance from the smart insoles or the Gait Up data compared with the actual data. ICC values are interpreted as follows: excellent (>0.90), good (0.75–0.90) moderate (0.50–0.74), and poor (<0.50) [63]. Since the sample size is small and the data distribution is non-normal, non-parametric statistical analyses were performed, particularly Mann–Whitney U tests. The Mann–Whitney U tests were used to determine whether there was a statistically significant difference between 1) Approach 1 and Approach 2 and 2) actual total distance and estimated distance from smart insoles and Gait Up. Data were analyzed using SPSS (version 29.0.2.0 (20); SPSS Inc., IBM, Chicago, IL, USA) and are reported as the median and interquartile. Statistical significance is defined as a *p*-value of less than 0.05.

## 3. Results

In this study, the actual stride length ranges from 1.327 m to 1.804 m, with a median [interquartile] of 1.591 [0.314] m at self-selected maximal safe walking speeds from 1.62 to 2.22 m/s (see Table 1 and Figure 7). An ICC of 0.977 (CI: 0.876–0.996) for the right side and 0.987 (CI: 0.878–0.996) for the left side are obtained for the stride length estimation using Gait Up monitors. For Approach 1 using smart insoles, the ICC of the stride length is 0.993 (CI: 0.846–0.999) and 0.995 (CI: 0.854–0.999) for the right and left sides, respectively. Similarly, ICC of 0.991 (CI: 0.726–0.999) and 0.993 (CI: 0.674–0.999) are reported for the right and left sides, respectively, using Approach 2 of the smart insoles for stride length estimation. The stride-length accuracies are presented in Table 2. There is no statistically significant difference between the two approaches using smart insoles for the stride length estimation (Mann–Whitney U test = 31.0, *p* = 0.959 for the right side, and U = 30.0, *p* = 0.878 for the left side) as illustrated in Table 3.

The actual total distance ranges from 541.4 m to 743.2 m with a median [interquartile] of 609.4 [199.4] meters (see Table 4 and Figure 8). The ICC for total distance is 0.980 (CI: 0.908–0.996) and 0.982 (CI: 0.917–0.996) for the right and left sides, respectively, when using the Gait Up monitors. The smart insoles based on Approach 1 yield an ICC of 0.992 (CI: 0.959–0.998) for the right side and 0.994 (CI: (0.972–0.999) for the left side. Furthermore, Approach 2 of the smart insoles provides an ICC of 0.996 (CI: 0.982–0.999) for the right side and 0.991 (CI: 0.941–0.998) for the left side. The total distance accuracy is listed in Table 5. There is no statistically significant difference between the actual distance and each estimated distance (Gait Up: U = 25.0; *p* = 0.505 (right), U = 27.0 *p* = 0.645 (left); Approach 1 of insole: U = 31.0; *p* = 0.959 (right), U = 26.0; *p* = 0.574 (left); Approach 2 of insole: U = 28.0; *p* = 0.721 (right) and U = 26.5; *p* = 0.599 (left)) as illustrated in Table 6.

## 4. Discussion

This study aimed to develop and validate novel model-based algorithms to estimate stride length and thus total distance using acceleration and pressure data integrated into a smart insole. Phase segmentation of the gait cycle was determined using heel-strike and toe-off events based on force sensors under the heel and great toe bones. The identification of toe-off and heel-strike events based on pressure data was consistent with that reported by Tirosh et al. [64]. In comparison to Suzuki’s study [39] which used only acceleration data, the use of pressure data enabled the detection of the toe-off event at the beginning of the antero–posterior acceleration increase. In Suzuki’s study [39], the toe-off event was detected as the instant of the first peak that appeared on the acceleration signal within the time interval between 0.1 and 0.4 s. The two selected approaches are based on the director coefficients of acceleration data (Approach 1) and dynamic time warping (Approach 2). The results show overall accuracy of over 97% with an ICC of 0.977 and higher for the stride length and the total distance using the Gait Up monitors. For Approach 1 of the smart insoles, the stride length, and total distance accuracies are greater than 98%, with an ICC of over 0.990. Accuracies of >98% with ICC > 0.991 are reported based on Approach 2 for stride length and total distance. There is no statistically significant difference between the reference distance and the estimated distance of either the Gait Up monitors or smart insoles. Altogether, the smart insoles and Gait Up monitors offer high accuracies for stride length and total distance estimation. In addition, the results of these two approaches of smart insoles were compared to those of the Gait Up monitors equipped with an IMU, and no statistically significant differences were observed.

In this study, the stride length is multiplied by the stride count to determine the total distance. However, the stride count used in this study yielded a mean accuracy of 99.90 ± 0.05%, resulting in an average error of 0.10% [52]. Another source of error in estimating total distance based on (1) can be derived from stride length estimation, as reported in this study. These two sources of error should be considered when using total distance estimation based on the formula (1) to interpret the results.

In the present study, the accuracies and ICC values of stride length and total distance are similar. These results show that although there was an error in stride counting, this error does not significantly affect the total distance estimation. Therefore, to determine the total distance that reflects the stride length, the stride count would need to be accurate. As Approach 2 was based on an algorithm and Approach 1 was based on a mathematical model, it was expected that Approach 2 would be more accurate. Contrary to our assumption that Approach 2 would be more accurate than Approach 1, the results of the two approaches are similar. This similarity could be due to the small sample size and the healthy participants with normal gaits.

In the literature, Pham et al. [37] reported distance accuracies of 98.89% to 99.26% using the conditional generative adversarial network-based regression approach and 98.57% using a deep neural network approach. Overall total distance accuracies based on conventional regression methods ranged from 95.88% to 97.91% across several studies [21,22,30,46,48,65]. Based on the gait models, accuracies of 97% [18] for stride length and 95.6% [19] for total distance were reported (see Table 7). For the gait models based on inverted pendulum models, an ICC of 0.89 (CI: 0.77- 0.95) [66] and ICC of 0.756 to 0.929 [67] were reported for stride length. The total distance accuracies based on direct method ranged from 90% to 97% [17,21,23,68,69]. Although several studies estimated stride length using approaches based on biomechanical, empirical, and parametric regression models, significant estimation errors remain, ranging from 3.4% to 16% [10,27,29,70,71,72,73]. The development of novel approaches such as the DTW-based model and the acceleration director coefficient-based model, contributes to reducing stride length estimation errors and, consequently, in the total distance covered. In fact, the approach based on the acceleration director coefficient yielded median errors of 1.08% to 1.2%, and the DTW-based approach had median error of 0.83% for stride length. Similarly, median errors in total distance covered 1.07% to 1.44% for the approach based on the acceleration director coefficient and 0.3% to 1.03% for the DTW-based approach were reported in this study. Zrenner et al. [10] used only the foot acceleration data to estimate stride length, whereas the present study used the foot pressure and acceleration data. In fact, the stride segmentation was based on the initial ground contacts of the acceleration data in their study [10] compared to our study where the stride segmentation was performed using pressure and acceleration data. Furthermore, Zrenner et al. [10] used a quadratic regression model to estimate velocity but the used approach to estimate stride length was not reported, while our study used the director coefficients of acceleration data and DTW to estimate the stride length. Gradl et al. [45] used a method that could be similar to those proposed in this paper. The difference between their method and our method lies in (1) the determination of the integration value (Gradl’s team used smoothing of the raw acceleration signal, whereas this study used the best fitting of straight line of acceleration signal) and (2) the variable of interest which is velocity [45].

In Rampp’s study [43], a mean absolute error of 6.26 ± 5.56 cm with Spearman correlation 0.93 in geriatric people (82.1 ± 6.5 years old) with normal walk. The study’s Mariani was reported mean errors of 0.4 ± 6.1% and 2.1 ± 6.8 with ICC of 0.91 (CI: 0.79–0.96) to estimating stride length in older people (mean age of 17.6 ± 4.6 years) and young people (mean age 26.1 ± 2.8 years), respectively [9]. No statistically significant difference between the young and older people for estimating stride length was reported [9].

This study had the following three limitations: the error source in the position trajectory of the reference model, during a 180° turn of the predefined pathway, and the use of a single axis of acceleration data to estimate stride length then losing information on rotation. In fact, the error of 4.6 cm appeared in the position/displacement trajectory during the time range of 0.0 s to 0.15 s (see Figure 9) when modelling the position trajectory as a third order polynomial. Another source of error can be derived from the displacement trajectory of the reference model, which was only considered for one participant’s data and differed from the path used by all the participants in this study. The total distance used as the gold standard did not consider the distance performed during a 180° turn, but only on 20 m straight line, yet the path was nearly oval, even with two parallel lines, as illustrated in Figure 1. Therefore, some distances are reduced to the actual total distance. Although the defined sensors positions are approximate, their positions in the interest zones can vary depending on participants’ morphological characteristics. The small sample size of this study does not enable the results to be generalized, although similar sample sizes are reported in the literature [69,74,75,76,77].

Further studies should consider the three axes of acceleration to avoid losing information on rotation when estimating stride length, and a gyroscope should be used in combination. Similarly, a motion capture system should be used as a reference system to estimate total distance covered in order to avoid systematic bias due to the reference distance (by multiplying round trips by 40 m). Before these approaches are used in real-world clinical practice, they will be assessed in people with gait impairments (stroke, Parkinson’s disease, frailty, etc.) to ensure the results can be generalized. Estimating the total distance covered is a parameter used to assess walking performance. Obtaining errors of approximately 1% in the estimation of the total distance covered could contribute to the use of smart insoles as a measurement tool when assessing walking performance in the future.

## 5. Conclusions

This study enabled the development and validation of two novel model-based approaches for measuring stride length for total distance estimation using a smart insole integrating accelerometer and pressure sensors in healthy participants. These two approaches are based on the director coefficients of the acceleration data (Approach 1) and dynamic time warping (Approach 2). Although the sample size was small and the reference system was not robust, the results showed that the accuracies and ICC values were very high for stride length and total distance estimations using both approaches. These approaches yielded similar results. Compared to model-based approaches in the literature, these two novel approaches appear to be more accurate for stride length and total distance estimation. However, further studies should be conducted, considering the limitations of this study.

## Figures and Tables

**Figure 1 sensors-25-06217-f001:**
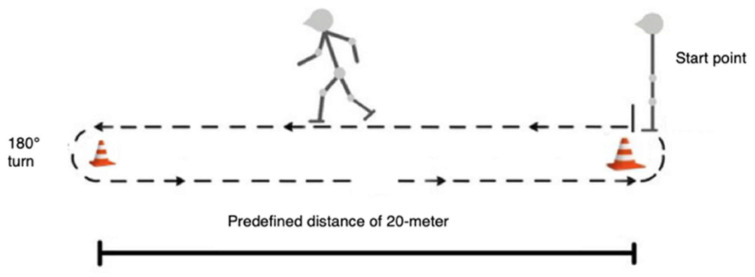
Pathway of walking performed by all participants.

**Figure 2 sensors-25-06217-f002:**
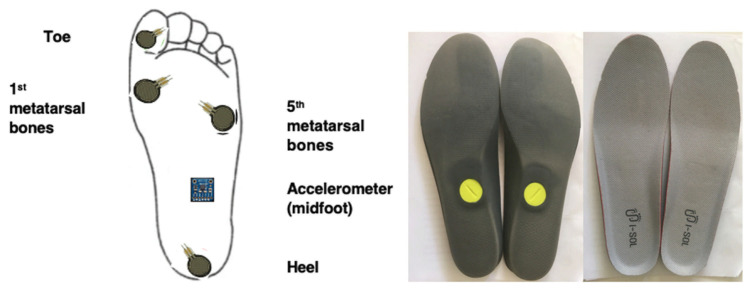
Sites of sensors under the smart insoles.

**Figure 3 sensors-25-06217-f003:**
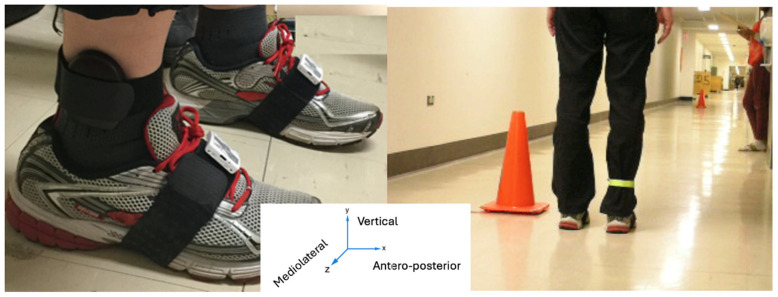
A pair of Gait Up monitors worn by a participant.

**Figure 4 sensors-25-06217-f004:**
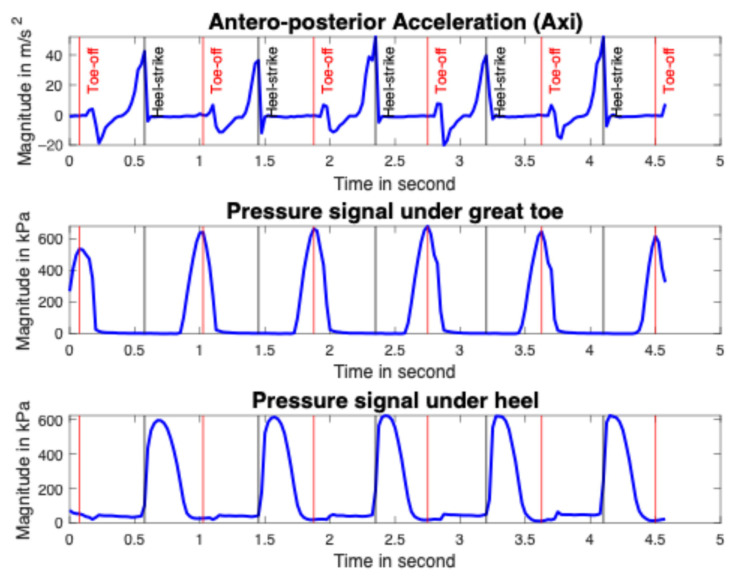
Identification of the heel-strike (in black) and toe-off (in red) events on acceleration and pressure (under heel and great toe) signals.

**Figure 5 sensors-25-06217-f005:**
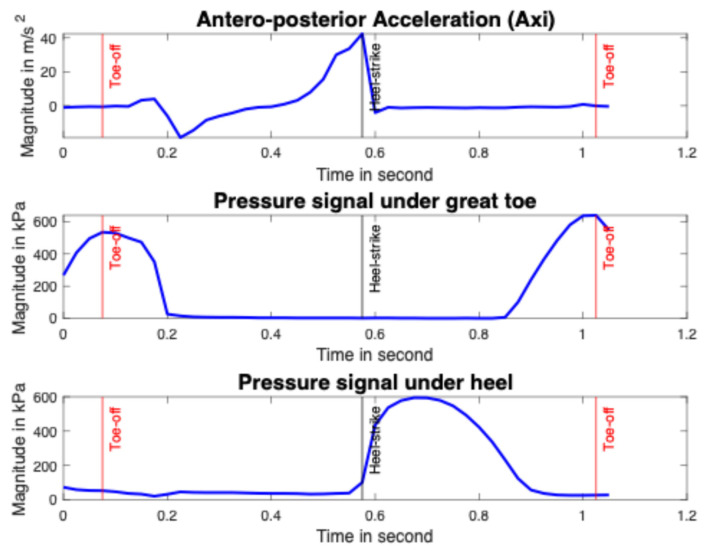
Signal identification of Figure 4 shows the toe-off and heel-strike events on a gait cycle.

**Figure 6 sensors-25-06217-f006:**
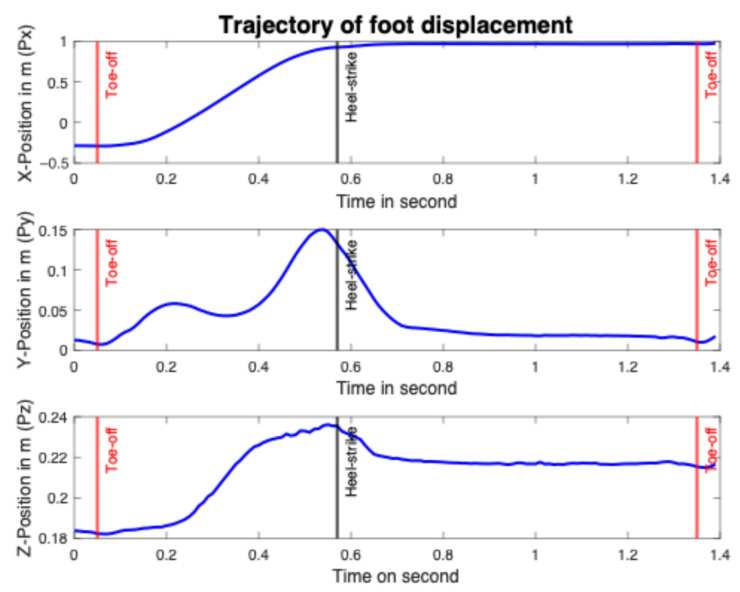
Identification of toe-off and heel-stride events on a gait cycle measured from Natural Point motion capture.

**Figure 7 sensors-25-06217-f007:**
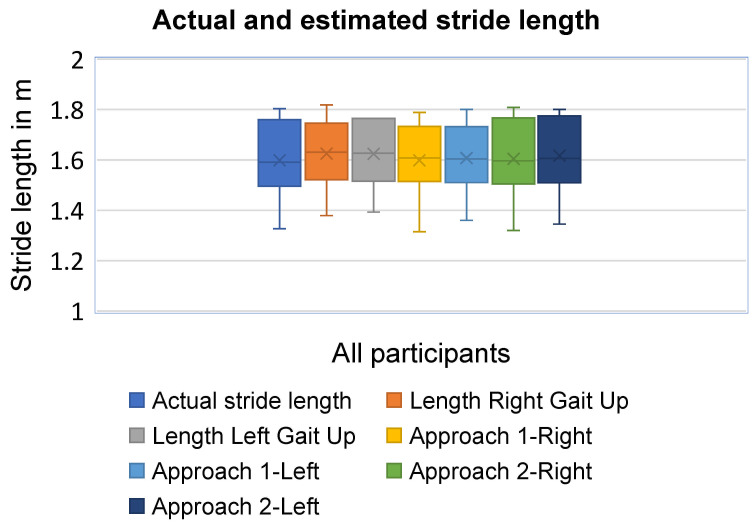
Stride lengths for gait up, smart insole, and actual stride length.

**Figure 8 sensors-25-06217-f008:**
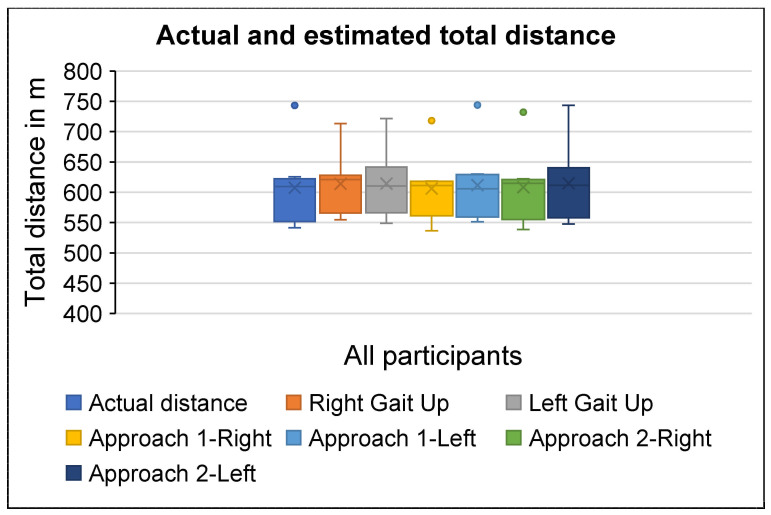
Total distance for Gait Up, smart insole, and actual total distance.

**Figure 9 sensors-25-06217-f009:**
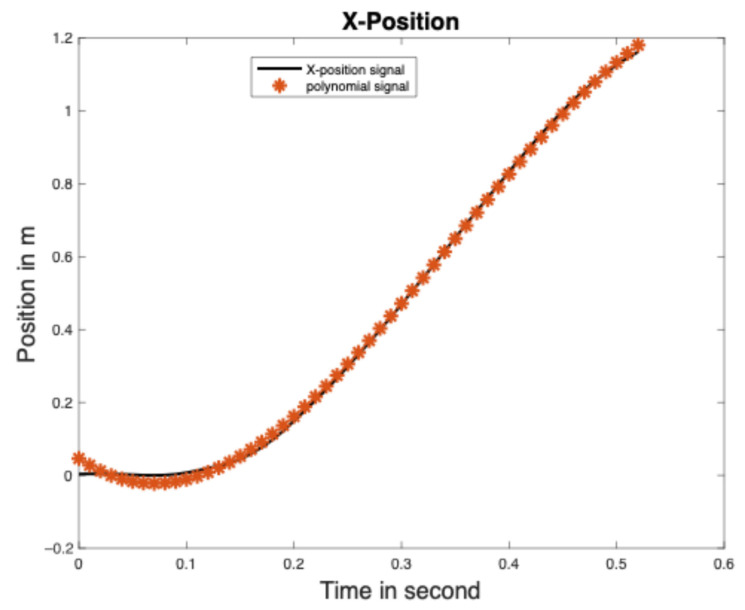
Representation of error source appeared in the position trajectory between polynomial signal and motion capture signal.

**Table 1 sensors-25-06217-t001:** Stride lengths of all participants estimated using Gait Up, smart insole, and actual stride length.

Participants	Actual Outcomes		Gait Up				Smart Insole					
	Number of Strides	Stride Length (m)	Number of Stride Right	Number of Stride Left	Stride Length Right (m)	Stride Length Left (m)	Number of Stride Right	Number of Stride Left	Approach 1-Stride Length Right (m)	Approach 1-Stride Length Left (m)	Approach 2-Stride Length Right (m)	Approach 2-Stride Length Left (m)
P1	368	1.66	368	368	1.698	1.685	367	368	1.681	1.682	1.668	1.673
P2	362	1.690	360	361	1.748	1.765	361	363	1.690	1.729	1.723	1.772
P3	365	1.490	363	365	1.535	1.504	364	366	1.522	1.506	1.503	1.503
P4	380	1.513	378	377	1.564	1.549	380	387	1.534	1.522	1.525	1.538
P5	341	1.783	340	340	1.819	1.765	342	342	1.789	1.733	1.808	1.775
P6	412	1.804	410	409	1.740	1.764	411	413	1.747	1.801	1.781	1.800
P7	411	1.522	411	410	1.517	1.569	409	413	1.512	1.525	1.510	1.529
P8	408	1.327	402	402	1.379	1.393	408	407	1.315	1.360	1.320	1.345
Median	374	1.591	373	372.5	1.631	1.627	373.5	377.5	1.608	1.604	1.597	1.606
Interquartile	49	0.293	50	48	0.231	0.261	48	50	0.235	0.227	0.278	0.272

**Table 2 sensors-25-06217-t002:** Stride length accuracies of smart insoles and Gait Up monitors compared to actual stride length.

	Stride Length Accuracies					
	Gait Up		Smart Insoles			
Participants	Right	Left	Approach 1-Right	Approach 1-Left	Approach 2-Right	Approach 2-Left
P1	97.77%	98.56%	98.92%	99.64%	98.68%	99.04%
P2	96.57%	95.56%	97.40%	97.10%	96.86%	96.63%
P3	96.98%	99.06%	97.58%	99.33%	97.38%	99.13%
P4	96.63%	97.62%	99.41%	98.48%	99.21%	98.28%
P5	97.98%	98.99%	98.88%	98.99%	98.54%	98.49%
P6	96.45%	97.78%	98.23%	99.28%	97.84%	99.06%
P7	99.67%	96.91%	97.77%	98.42%	97.63%	98.23%
P8	96.08%	95.03%	99.02%	99.40%	99.25%	99.25%
Median	96.80%	97.70%	98.92%	98.80%	99.17%	99.17%
Interquartile	1.32%	3.43%	1.44%	2.30%	2.35%	2.50%

**Table 3 sensors-25-06217-t003:** Intraclass correlation coefficients and Mann–Whitney U test for stride length.

Sides	For Stride Length			
	Intraclass Correlation Coefficient (CI)			Mann–Whitney U Test
	Smart Insole		Gait Up	Smart Insole
	Approach 1	Approach 2		Approach 1 and Approach 2
**Right**	ICC = 0.993 (0.846–0.999	ICC = 0.991 (0.726–0.999)	ICC = 0.977 (0.876–0.996)	U = 31.0 (*p* = 0.959)
**Left**	ICC = 0.995 (0.854–0.999)	ICC = 0.993 (0.674–0.999)	ICC = 0.987 (0.878–0.996)	U = 30.0 (*p* = 0.878)

CI: confidence interval; *p*: *p*-value.

**Table 4 sensors-25-06217-t004:** Total distance covered in meters of all participants using Gait Up, smart insole, and actual total distance.

Partici- pants (P)	Actual Distance (m)	Gait Up		Smart Insole			
		Right	Left	Approach 1-Right	Approach 1-Left	Approach 2-Right	Approach 2-Left
P1	610.88	624.86	620.08	616.93	618.98	612.16	615.66
P2	611.78	629.28	637.17	610.09	627.63	622.00	643.24
P3	543.85	557.21	548.96	554.01	551.20	547.09	550.10
P4	574.94	591.19	583.97	582.92	575.32	579.5	581.36
P5	608.00	618.46	600.1	611.84	592.69	618.34	607.05
P6	743.25	713.4	721.48	718.02	743.81	731.99	743.4
P7	625.54	623.49	643.29	618.41	629.83	617.59	631.48
P8	541.42	554.36	559.99	536.52	553.52	538.56	547.42
Median	609.44	620.97	610.09	610.96	605.83	614.87	611.36
Interquartile	81.69	66.28	94.33	64.4	78.63	70.50	81.38

**Table 5 sensors-25-06217-t005:** Total distance accuracies of the smart insole and Gait Up monitors compared to actual.

	Total Distance Accuracies					
	Gait Up		Smart Insoles			
Participants	Right	Left	Approach 1-Right	Approach 1-Left	Approach 2-Right	Approach 2-Left
P1	97.71%	98.49%	99.01%	98.67%	99.79%	99.22%
P2	97.14%	95.85%	99.72%	97.41%	98.33%	94.86%
P3	97.54%	99.06%	98.13%	98.65%	99.40%	98.85%
P4	97.17%	98.43%	98.61%	99.93%	99.21%	98.88%
P5	98.28%	98.70%	99.37%	97.48%	98.30%	99.84%
P6	95.98%	97.07%	96.61%	99.92%	98.49%	99.98%
P7	99.67%	97.16%	98.86%	99.32%	98.73%	99.05%
P8	97.61%	96.57%	99.10%	97.76%	99.47%	98.89%
Median	97.58%	97.80%	98.93%	98.66%	98.97%	99.97%
Interquartile	1.14%	2,13%	1.24%	2.44%	0.18%	0.14%

**Table 6 sensors-25-06217-t006:** Intraclass correlation coefficients and Mann–Whitney U test for total distance covered.

Sides	Total Distance Covered					
	Intraclass Correlation Coefficient (CI)			Mann–Whitney U Test		
	Smart Insole		Gait Up	Smart Insole		Gait Up
	Approach 1	Approach 2		Approach 1	Approach 2	
**Right**	ICC = 0.992 (0.959–0.998)	ICC = 0.996 (0.982–0.999)	ICC = 0.980 (0.908–0.996	U = 31.0; *p* = 0.959	U = 28.0; *p* = 0.721	U = 25.0; *p* = 0.505
**Left**	ICC = 0.994 (0.972–0.999)	ICC = 0.991 (0.941–0.998)	ICC = 0.982 (0.917–0.996)	U = 26.0; *p* = 0.574	U = 26.5; *p* = 0.599	U = 27.0; *p* = 0.645

CI: confidence interval; *p*: *p*-value.

**Table 7 sensors-25-06217-t007:** Comparison of gait model-based approaches accuracies.

Author	Approaches	Results
Bennett et al. [18]	Modeling human leg as a two-link revolute robot, then using extended Kalman filter (EKF) to estimate the displacement in a straight line	Accuracy of 97% on linear displacement
Meng et al. [19]	Self-contained pedestrian tracking approach using a foot-mounted inertial and magnetic sensor module with traditional zero velocity updates and stride information to further correct the acceleration double integration drifts	For short distance: position error of 0.44 ± 0.20 m on straight line of 10 m and 15 m, 0.45 ± 0.08 m on 180° turn, and 0.40 ± 0.07 m on circle of radius 3 m For long distance of 3 min indoor walking, position error of 4.33 ± 1.77 m and 3.88 ± 0.35 m for 6 min outdoor walking
Miyazaki et al. [27]	Simplified two-segment leg model and implemented a stride length estimation algorithm based on a single gyroscope	Relative estimation error of 15%
Kim et al. [29]	Analyzing the relationship between stride, step period, and acceleration.	Calculated accuracies of 96.59% (error: 3.40%) for 1st test and 95.83 (error: 4.17%) for 2nd test
Xia et al. [31]	A non-linear step length estimation model based on statistics proposed by [30] where Step = k·^4^√αz − max − αz–min	Not reported
Aminian et al. [70]	Stride length estimation using sensor units at thigh and shank in combination with a double-inverted pendulum model	Root mean square error (RMSE) of 7.2% (0.07 m)
Zijlstra et al. [73]	Inverted pendulum to model the center of mass (CoM) trajectory	86% to 94% of accuracy for stride length
Zijlstra et al. [34]	Empirical approximation of the geometric model to estimate the stride length due to changes of vertical displacement of the CoM	Underestimation of stride length for all participants
Gonzalez et al. [71]	Model by extracting additional features from the acceleration signal to further improve the step length estimation	Stride length error of 16%
Lueken et al. [72]	Model of the inverted pendulum and idea of double integration of the antero–posterior acceleration	Stride length error from 8.52% to 12.87%
Lueken et al. [72]	Kalman filter and idea of double integration of the antero–posterior acceleration	Stride length error from 7.95% to 15.00%
Zrenner et al. [10]	Parametric regression model based on acceleration	Stride length error 7.9%
Ziagskas et al. [13]	Artificial intelligence algorithms	ICC = 0.939 for left side and ICC = 0.939 for right side
Our Approach 1	Model based on director coefficients of acceleration data	For stride length, right: 98.92% (error: 1.08%) and left: 98.80% (error: 1.20%) For total distance, right: 98.9 (3.10)% (error: 1.08%) and left: 98.69 (2.50%) (error: 1.31%)
Our Approach 2	Model based on the dynamic time warping	For stride length, right: 99.17% (error: 0.83%) and left: 98.17% (error: 0.83%) For total distance, right: 98.95 (0.10)% (error: 1.05%) and left: 99.03 (5.10)% (error: 0.97%)

## Data Availability

The data are published and available at https://doi.org/10.21227/fy4e-5220.

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
