# Peer review of "Validation of Novel Stride Length Model-Based Approaches to Estimate Distance Covered Based on Acceleration and Pressure Data During Walking"

_sensors, 2025, doi:10.3390/s25196217_

Round 1
Reviewer 1 Report
Comments and Suggestions for Authors
- The Abstract mixes “median accuracies” for Approach-1 with “average accuracies” for Approach-2, and even shows a negative “±” value (e.g., 98.92 ± **–**3.10%). Please use a consistent summary statistic (median [IQR] vs mean ± SD) across approaches and remove impossible signs. Also, “differences” is singular/plural inconsistent (“no statistically significant differences”).
- The “actual distance” excludes the 180° turns (noted later in Limitations), yet round trips are “calculated directly for a single foot”. Please define operationally how a “single-foot round trip” is detected and counted, how partial laps within 6 minutes were handled, and reconcile this with the later acknowledgment that the path is effectively oval.
- Smart-insoles sample acceleration/pressure at 40 Hz, while Gait Up devices are separate units; the manuscript does not explain time alignment/synchronization between device clocks, nor any resampling/latency handling. Please describe how signals were synchronized during segmentation and algorithm evaluation.
- The text states heel-strike “matches with the maximum horizontal acceleration peak”, yet the algorithm description says heel-strike is detected using “islocalmin” on horizontal acceleration (a minimum). Resolve this contradiction and specify the exact segmentation rule.
- As written, (3) combines a scalar polynomial with a 3-vector without clarifying that each c_j is a 3-vector (or that (3) holds componentwise). Please restate (3)–(5) cleanly (vector form or per-axis form) and define the “director coefficients” explicitly (units, axes, and how the line fit is obtained).
- Results text reports ICCs and Mann–Whitney U tests appropriately, but please ensure every claim (e.g., “no statistically significant difference” between estimates and actual) is accompanied by exact U and p values in the same paragraph (you have most of them—finish aligning where missing or summarize them once in a coherent table).
- In Table 2 and Table 4 the “Inter-quartile” line includes negative values (e.g., –0.66%, –3.12%). By definition, an IQR is non-negative; if the intent is something else (e.g., median change), please rename and define precisely. Otherwise correct the signs.
- The authors state three limitations (polynomial modeling error near 0–0.15 s, path geometry/turns vs. gold standard, and single-axis usage). Please quantify (where possible) how each limitation might bias accuracies/ICCs and indicate how future work will mitigate them (e.g., tri-axial use, turn-inclusion policy, multi-participant reference trajectories).
Author Response
For research article
|
Response to Reviewer 1 Comments
|
||||||||||||||||||||||||||||||||||||||||||||||||||||||||||||||||||||||||||||||||||||||||||||||||||||||||||||||||||||||||||||||||||||||||||||||||||||||||||||||||||||||||||||||||||||||||||||||||||||||||||||||||||||||||||||||||||||||||||||||||||||||||||||||||||||||||||||||||||||||||||||||||||||||||||||||||||||||||||||||||||||||||||||||||||||||||||||||||||||||||||||||||||||||||||||||||||||||||||||||||||||||
|
1. Summary |
|
|
||||||||||||||||||||||||||||||||||||||||||||||||||||||||||||||||||||||||||||||||||||||||||||||||||||||||||||||||||||||||||||||||||||||||||||||||||||||||||||||||||||||||||||||||||||||||||||||||||||||||||||||||||||||||||||||||||||||||||||||||||||||||||||||||||||||||||||||||||||||||||||||||||||||||||||||||||||||||||||||||||||||||||||||||||||||||||||||||||||||||||||||||||||||||||||||||||||||||||||||||||||
|
The authors thank you for taking the time to review this manuscript. Your comments have helped us improve our manuscript. We have addressed each of your comments in our responses.
|
||||||||||||||||||||||||||||||||||||||||||||||||||||||||||||||||||||||||||||||||||||||||||||||||||||||||||||||||||||||||||||||||||||||||||||||||||||||||||||||||||||||||||||||||||||||||||||||||||||||||||||||||||||||||||||||||||||||||||||||||||||||||||||||||||||||||||||||||||||||||||||||||||||||||||||||||||||||||||||||||||||||||||||||||||||||||||||||||||||||||||||||||||||||||||||||||||||||||||||||||||||||
|
2. Questions for General Evaluation |
Reviewer’s Evaluation |
Response and Revisions |
||||||||||||||||||||||||||||||||||||||||||||||||||||||||||||||||||||||||||||||||||||||||||||||||||||||||||||||||||||||||||||||||||||||||||||||||||||||||||||||||||||||||||||||||||||||||||||||||||||||||||||||||||||||||||||||||||||||||||||||||||||||||||||||||||||||||||||||||||||||||||||||||||||||||||||||||||||||||||||||||||||||||||||||||||||||||||||||||||||||||||||||||||||||||||||||||||||||||||||||||||||
|
Does the introduction provide sufficient background and include all relevant references? |
Yes/Can be improved/Must be improved/Not applicable |
|
||||||||||||||||||||||||||||||||||||||||||||||||||||||||||||||||||||||||||||||||||||||||||||||||||||||||||||||||||||||||||||||||||||||||||||||||||||||||||||||||||||||||||||||||||||||||||||||||||||||||||||||||||||||||||||||||||||||||||||||||||||||||||||||||||||||||||||||||||||||||||||||||||||||||||||||||||||||||||||||||||||||||||||||||||||||||||||||||||||||||||||||||||||||||||||||||||||||||||||||||||||
|
Are all the cited references relevant to the research? |
Yes/Can be improved/Must be improved/Not applicable |
|
||||||||||||||||||||||||||||||||||||||||||||||||||||||||||||||||||||||||||||||||||||||||||||||||||||||||||||||||||||||||||||||||||||||||||||||||||||||||||||||||||||||||||||||||||||||||||||||||||||||||||||||||||||||||||||||||||||||||||||||||||||||||||||||||||||||||||||||||||||||||||||||||||||||||||||||||||||||||||||||||||||||||||||||||||||||||||||||||||||||||||||||||||||||||||||||||||||||||||||||||||||
|
Is the research design appropriate? |
Yes/Can be improved/Must be improved/Not applicable |
|
||||||||||||||||||||||||||||||||||||||||||||||||||||||||||||||||||||||||||||||||||||||||||||||||||||||||||||||||||||||||||||||||||||||||||||||||||||||||||||||||||||||||||||||||||||||||||||||||||||||||||||||||||||||||||||||||||||||||||||||||||||||||||||||||||||||||||||||||||||||||||||||||||||||||||||||||||||||||||||||||||||||||||||||||||||||||||||||||||||||||||||||||||||||||||||||||||||||||||||||||||||
|
Are the methods adequately described? |
Yes/Can be improved/Must be improved/Not applicable |
|
||||||||||||||||||||||||||||||||||||||||||||||||||||||||||||||||||||||||||||||||||||||||||||||||||||||||||||||||||||||||||||||||||||||||||||||||||||||||||||||||||||||||||||||||||||||||||||||||||||||||||||||||||||||||||||||||||||||||||||||||||||||||||||||||||||||||||||||||||||||||||||||||||||||||||||||||||||||||||||||||||||||||||||||||||||||||||||||||||||||||||||||||||||||||||||||||||||||||||||||||||||
|
Are the results clearly presented? |
Yes/Can be improved/Must be improved/Not applicable |
|
||||||||||||||||||||||||||||||||||||||||||||||||||||||||||||||||||||||||||||||||||||||||||||||||||||||||||||||||||||||||||||||||||||||||||||||||||||||||||||||||||||||||||||||||||||||||||||||||||||||||||||||||||||||||||||||||||||||||||||||||||||||||||||||||||||||||||||||||||||||||||||||||||||||||||||||||||||||||||||||||||||||||||||||||||||||||||||||||||||||||||||||||||||||||||||||||||||||||||||||||||||
|
Are the conclusions supported by the results? |
Yes/Can be improved/Must be improved/Not applicable |
|
||||||||||||||||||||||||||||||||||||||||||||||||||||||||||||||||||||||||||||||||||||||||||||||||||||||||||||||||||||||||||||||||||||||||||||||||||||||||||||||||||||||||||||||||||||||||||||||||||||||||||||||||||||||||||||||||||||||||||||||||||||||||||||||||||||||||||||||||||||||||||||||||||||||||||||||||||||||||||||||||||||||||||||||||||||||||||||||||||||||||||||||||||||||||||||||||||||||||||||||||||||
|
3. Point-by-point response to Comments and Suggestions for Authors |
||||||||||||||||||||||||||||||||||||||||||||||||||||||||||||||||||||||||||||||||||||||||||||||||||||||||||||||||||||||||||||||||||||||||||||||||||||||||||||||||||||||||||||||||||||||||||||||||||||||||||||||||||||||||||||||||||||||||||||||||||||||||||||||||||||||||||||||||||||||||||||||||||||||||||||||||||||||||||||||||||||||||||||||||||||||||||||||||||||||||||||||||||||||||||||||||||||||||||||||||||||||
|
Comments 1: The Abstract mixes “median accuracies” for Approach-1 with “average accuracies” for Approach-2, and even shows a negative “±” value (e.g., 98.92 ± **–**3.10%). Please use a consistent summary statistic (median [IQR] vs mean ± SD) across approaches and remove impossible signs. Also, “differences” is singular/plural inconsistent (“no statistically significant differences”).
|
||||||||||||||||||||||||||||||||||||||||||||||||||||||||||||||||||||||||||||||||||||||||||||||||||||||||||||||||||||||||||||||||||||||||||||||||||||||||||||||||||||||||||||||||||||||||||||||||||||||||||||||||||||||||||||||||||||||||||||||||||||||||||||||||||||||||||||||||||||||||||||||||||||||||||||||||||||||||||||||||||||||||||||||||||||||||||||||||||||||||||||||||||||||||||||||||||||||||||||||||||||||
|
Response 1: Thank you for your constructive comments. The corrections have been made in the manuscript: “The median accuracies of the total distance using Approach-1 are 98.92% [1.24%] (ICC=0.992) and 98.69% [2.44%] (ICC=0.994) for the right and left sides, respectively. For Approach-2, the average accuracies are 98.95% [0.18%] (ICC=0.996) for the right side and 99.03% [0.14%] (ICC=0.991) for the left side.” Please see Page 1, line 26 to 28 and “In this study, the actual stride length ranges from 1.327 m to 1.804 m, with a median [interquartile] of 1.591 [0.314] m at self-selected maximal safe walking speeds from 1.62 to 2.22 m/s (see Table 1 and Fig. 7). “ ….” The actual total distance ranges from 541.4 meters to 743.2 meters with a median [interquartile] of 609.4 [199.4] meters (see Table 3 and Fig. 8). “See page 8, lines 261, 272 and 273. |
||||||||||||||||||||||||||||||||||||||||||||||||||||||||||||||||||||||||||||||||||||||||||||||||||||||||||||||||||||||||||||||||||||||||||||||||||||||||||||||||||||||||||||||||||||||||||||||||||||||||||||||||||||||||||||||||||||||||||||||||||||||||||||||||||||||||||||||||||||||||||||||||||||||||||||||||||||||||||||||||||||||||||||||||||||||||||||||||||||||||||||||||||||||||||||||||||||||||||||||||||||||
|
Comments 2: The “actual distance” excludes the 180° turns (noted later in Limitations), yet round trips are “calculated directly for a single foot”. Please define operationally how a “single-foot round trip” is detected and counted, how partial laps within 6 minutes were handled, and reconcile this with the later acknowledgment that the path is effectively oval.
|
||||||||||||||||||||||||||||||||||||||||||||||||||||||||||||||||||||||||||||||||||||||||||||||||||||||||||||||||||||||||||||||||||||||||||||||||||||||||||||||||||||||||||||||||||||||||||||||||||||||||||||||||||||||||||||||||||||||||||||||||||||||||||||||||||||||||||||||||||||||||||||||||||||||||||||||||||||||||||||||||||||||||||||||||||||||||||||||||||||||||||||||||||||||||||||||||||||||||||||||||||||||
|
Response 2: In data collection section, the description has been added in the manuscript: “Each participant indicated in advance which foot they would use to start walking, and fluorescent Velcro was attached to the shin of that foot. During the walk, team members incremented the manual counter each time the participant's Velcro-equipped foot took a stride. For each participant, the total distance was calculated by multiplying the number of round trips by 40 m for a single foot and was used as the actual distance. Once 6-MWT had elapsed, the participant stopped, and a team member measured the distance between the cone and the participant using an odometer. Similarly, if a half turn had also been completed, this was also added. These distances were added to the number of round trips multiplied by 40 m to determine the total distance covered. » See page 3, lines 125 to 128 and 130 to 134.
Comments 3: Smart-insoles sample acceleration/pressure at 40 Hz, while Gait Up devices are separate units; the manuscript does not explain time alignment/synchronization between device clocks, nor any resampling/latency handling. Please describe how signals were synchronized during segmentation and algorithm evaluation.
Response 3: For synchronization, the smart insole and Gait Up monitor were reset at the same time. The information has been added in data collection section. See Page 121 and 122.
Comments 4: The text states heel-strike “matches with the maximum horizontal acceleration peak”, yet the algorithm description says heel-strike is detected using “islocalmin” on horizontal acceleration (a minimum). Resolve this contradiction and specify the exact segmentation rule. Response 4: The correction has been added in the manuscript: “For identification of toe-off events, the function “findpeaks” in the MATLAB is used on the pressure under the great toe. The function “islocalmax” is used on the antero-posterior acceleration data for the identification of heel-strike. In fact, the function “islocalmax” returns a logical array, whose elements are 1 (true) when a local maximum is detected in the corresponding element of the acceleration.” see page 5, lines 170 to 174. Comments 5: As written, (3) combines a scalar polynomial with a 3-vector without clarifying that each c_j is a 3-vector (or that (3) holds componentwise). Please restate (3)–(5) cleanly (vector form or per-axis form) and define the “director coefficients” explicitly (units, axes, and how the line fit is obtained). Response 5: The correction has been added on page 7 in the manuscript, page 7 line 207 to 223: “P3 x nm(t) = c1t3 + c2t2 + c3t + c4 = [Pmx Pmy Pmz]T (3) where cj is a vector of constants (with 1<=j=<4) to find, t is the time over a window (segmented signal of the model) of n samples, T is the period of the corresponding stride cycle defined by two adjacent toe-off events and Pm(t) is the resulting position model. The first and second derivative make it possible to find the velocity Vm(t), described in (4) and the acceleration Am(t) described in (5) respectively. The representation of Vm(t) as a second-order polynomial was reported in some studies [10, 45]. V3 x nm (t) = 3c1t2 + 2c2t + c3 = [Vmx Vmy Vmz]T (4) A3 x nm (t) = 6c1t + 2c2 = [Amx Amy Amz]T (5) … From these equations (3) to (5), only x-axis (anterio-posterior axis) is used Pmx, Vmx and Amx. For the Approach-1, the ratio is calculated with the director coefficients of the antero-posterior acceleration of the smart insole (Asx(t), using subscript s for describing the signal acquired from the smart insole) and the reference model (Amx) illustrated in (5). The director coefficient of the reference model is 6c1. For smart insole, the straight line that best fits with the swing phase data (Asxn(t)) for n samples was determined based on the antero-posterior acceleration (Asx(t)) for each gait cycle. The “lsqcurvefit” function was used todetermine the director coefficient of Asxn(t)) for each gait cycle. The director co-efficients have no units. This ratio (R) is then used to estimate the average stride length (L), as illustrated in the Algorithm 1.” Comments 6: Results text reports ICCs and Mann–Whitney U tests appropriately, but please ensure every claim (e.g., “no statistically significant difference” between estimates and actual) is accompanied by exact U and p values in the same paragraph (you have most of them—finish aligning where missing or summarize them once in a coherent table).
Response 6: The correction have been added in manuscript, lines 268 to 271; lines 279 to 282 and 297 to 301 : “There is no statistically significant difference between the two approaches using smart insoles for the stride length estimation (Mann-Whitney U test= 31.0, p=0.959 for the right side and U= 30.0, p=0.878 for the left side) as illustrated in Table 5”. “There is no statistically significant difference between the actual distance and each estimated distance (Gait Up: U=25.0; p=0.505 (right), U=27.0 p=0.645 (left); Approach-1 of insole: U=31.0; p=0.959 (right), U=26.0; p=0.574 (left); Approach-2 of insole: U=28.0; p=0.721 (right) and U=26.5; p=0.599 (left)) as illustrated in Table 6.” The table 5 and table 6 were added in the manuscript. Table 5. Intraclass correlation coefficients and Mann-Whitney U test for Stride length
CI: confidence interval; p: p-value Table 6: Intraclass correlation coefficients and Mann-Whitney U test for total distance covered
CI: confidence interval; p: p-value
Comments 7: In Table 2 and Table 4 the “Inter-quartile” line includes negative values (e.g., –0.66%, –3.12%). By definition, an IQR is non-negative; if the intent is something else (e.g., median change), please rename and define precisely. Otherwise correct the signs.
Response 7: Corrections have been made on lines 285 to 288, 291 to 293 and 296. Table 1. Strides length of all participants estimated using Gait Up, smart insole and actual stride length
Table 2. Stride length accuracies of smart insoles and Gait Up monitors compared to actual stride length
Table 4. Total distance accuracies of the smart insole and Gait Up monitors compared to actual
Comments 8: The authors state three limitations (polynomial modeling error near 0–0.15 s, path geometry/turns vs. gold standard, and single-axis usage). Please quantify (where possible) how each limitation might bias accuracies/ICCs and indicate how future work will mitigate them (e.g., tri-axial use, turn-inclusion policy, multi-participant reference trajectories).
Response 8: the error of 4.6 cm appeared in the position/displacement trajectory during the time range of 0.0 s to 0.15 seconds (see figure 9) when modelling the position trajectory as a third-order polynomial. However, for further studies, the three axes of acceleration should be considered in order to avoid losing information on rotation when estimating stride length, and a gyroscope should be used in combination. Similarly, a motion capture system should be used as a reference system to estimate total distance covered. Pages 15 and 16.
Figure 9: Representation of error source appeared in the position trajectory between polynomial signal and motion capture signal
|
||||||||||||||||||||||||||||||||||||||||||||||||||||||||||||||||||||||||||||||||||||||||||||||||||||||||||||||||||||||||||||||||||||||||||||||||||||||||||||||||||||||||||||||||||||||||||||||||||||||||||||||||||||||||||||||||||||||||||||||||||||||||||||||||||||||||||||||||||||||||||||||||||||||||||||||||||||||||||||||||||||||||||||||||||||||||||||||||||||||||||||||||||||||||||||||||||||||||||||||||||||||
|
4. Response to Comments on the Quality of English Language |
||||||||||||||||||||||||||||||||||||||||||||||||||||||||||||||||||||||||||||||||||||||||||||||||||||||||||||||||||||||||||||||||||||||||||||||||||||||||||||||||||||||||||||||||||||||||||||||||||||||||||||||||||||||||||||||||||||||||||||||||||||||||||||||||||||||||||||||||||||||||||||||||||||||||||||||||||||||||||||||||||||||||||||||||||||||||||||||||||||||||||||||||||||||||||||||||||||||||||||||||||||||
|
Point 1: |
||||||||||||||||||||||||||||||||||||||||||||||||||||||||||||||||||||||||||||||||||||||||||||||||||||||||||||||||||||||||||||||||||||||||||||||||||||||||||||||||||||||||||||||||||||||||||||||||||||||||||||||||||||||||||||||||||||||||||||||||||||||||||||||||||||||||||||||||||||||||||||||||||||||||||||||||||||||||||||||||||||||||||||||||||||||||||||||||||||||||||||||||||||||||||||||||||||||||||||||||||||||
|
Response 1: The quality of English Language was improved in the manuscript in red. |
||||||||||||||||||||||||||||||||||||||||||||||||||||||||||||||||||||||||||||||||||||||||||||||||||||||||||||||||||||||||||||||||||||||||||||||||||||||||||||||||||||||||||||||||||||||||||||||||||||||||||||||||||||||||||||||||||||||||||||||||||||||||||||||||||||||||||||||||||||||||||||||||||||||||||||||||||||||||||||||||||||||||||||||||||||||||||||||||||||||||||||||||||||||||||||||||||||||||||||||||||||||
|
5. Additional clarifications |
||||||||||||||||||||||||||||||||||||||||||||||||||||||||||||||||||||||||||||||||||||||||||||||||||||||||||||||||||||||||||||||||||||||||||||||||||||||||||||||||||||||||||||||||||||||||||||||||||||||||||||||||||||||||||||||||||||||||||||||||||||||||||||||||||||||||||||||||||||||||||||||||||||||||||||||||||||||||||||||||||||||||||||||||||||||||||||||||||||||||||||||||||||||||||||||||||||||||||||||||||||||
|
[Here, mention any other clarifications you would like to provide to the journal editor/reviewer.] |
||||||||||||||||||||||||||||||||||||||||||||||||||||||||||||||||||||||||||||||||||||||||||||||||||||||||||||||||||||||||||||||||||||||||||||||||||||||||||||||||||||||||||||||||||||||||||||||||||||||||||||||||||||||||||||||||||||||||||||||||||||||||||||||||||||||||||||||||||||||||||||||||||||||||||||||||||||||||||||||||||||||||||||||||||||||||||||||||||||||||||||||||||||||||||||||||||||||||||||||||||||||

Reviewer 2 Report
Comments and Suggestions for Authors
The study addresses the estimation of stride length and total distance covered through wearable sensors. Accurate, unobtrusive gait assessment tools are highly valuable in clinical rehabilitation, sports science, and public health. The paper introduces two new model-based approaches (director coefficients of acceleration data and Dynamic Time Warping), expanding beyond conventional accelerometer-only solutions. The manuscript is generally well structured and easy to read.
However, there are several major concerns. Some of these are acknowledged by the authors as limitations, but they represent serious methodological weaknesses.
Major concerns
- The study includes only eight healthy participants, a number far too small to validate a new model, especially considering age, gait variability, and pathological conditions that would represent actual clinical populations. The absence of participants with gait impairments (stroke, Parkinson’s disease, frailty, etc.) limits the applicability of the findings to real-world clinical practice.
- The reference (“multiplying round trips × 40 m”) ignores turning maneuvers and the geometry of the walking path. This introduces a systematic bias that is openly acknowledged but not properly corrected.
- The model is derived from a single participant’s motion capture data, which raises questions about representativeness and robustness.
- On what basis is the initial hypothesis made that the DTW methodology would be more accurate than the approach based on director coefficients of acceleration data? This requires proper justification.
- A clearer justification of the novelty and added value of the proposed algorithms is needed.
- Statistical power is insufficient, and the conclusions about “excellent” validity are overstated. The reliance on ICCs with very narrow confidence intervals seems overly optimistic given the small sample size. Non-parametric tests (Mann-Whitney) are used, but the choice is not well justified, nor is there any adjustment for multiple comparisons.
- 7. The discussion is very limited and should be expanded to cover at least (but not only) clinical aspects and the interpretation of results. While technical accuracy is reported, clinical utility is not demonstrated. Would these methods provide meaningful improvements in patient monitoring compared to existing commercial solutions (e.g., IMU-based gait monitors)? The study does not define a minimal clinically important difference (MCID) for distance estimation, making it unclear whether a 1–2% gain in accuracy has any practical relevance.
Despite proposing two novel approaches, the study concludes that both perform similarly, yet no strong rationale is provided for why this similarity occurs or why DTW—despite being more computationally demanding—should be preferred. Moreover, the algorithms were tested only on healthy gait patterns with relatively consistent strides, which likely inflates the performance metrics. Discuss differ results can be expected in a pathological population.
Minor concerns
- L59: The results of reference 13 are quite good; I would not include them in this sentence. There is also confusion between error and accuracy throughout the manuscript. The authors should pay attention not to conflate these two measures.
- L114: Please include information about the producer the first time the sensors are mentioned.
- L116: Walking speed range is a result; it should be included in the Results section, not in Materials and Methods.
- L120: The “gold standard” for total distance (multiplying round trips × 40 m) ignores turning maneuvers and the geometry of the walking path. This introduces a systematic bias that is acknowledged but not properly corrected.
- L130: Please provide more details when describing the smart insoles. Report technical characteristics of the embedded sensors and the insole material or provide a reference paper. Are the insoles self-made or commercial?
- L149: You mention horizontal acceleration. This is not a good description; I assume you mean antero-posterior acceleration. In the figures, acceleration along the x-axis is reported, but the reference system is never defined. I suggest including a reference system in Figure 1 or 3 and replacing the term horizontal acceleration with antero-posterior acceleration.
- L154–160: It is not useful to list the specific Matlab functions. A description of the implemented algorithm is sufficient.
- Figures 4 and 5: Gait events are indicated, but it is unclear where toe-offs or heel-strikes occur. Please mark them directly on the graph. Measurement units for the pressure signal are missing. The caption of Figure 4 should read identification rather than segmentation. In Figure 5, you claim to show a gait cycle, but the trend does not appear to represent a complete cycle. In particular, the initial and final pressure values are very different.
- L167–168: “Six markers were positioned on the front and back feet of a participant.” This is too generic; please specify the anatomical landmarks where the markers were placed.
- Figure 6: Same issues as Figures 4 and 5—unclear whether this represents a complete cycle, missing measurement units, unclear identification of heel-strike and toe-off. The graphs represent marker positions, but which marker? And according to which reference system (not defined)?
- L240–242: The number of participants, demographic, and anthropometric data should be reported in Materials and Methods, including sex/gender.
- Table 1: Units of measurement are missing. There is also confusion between stride and step.
The manuscript falls short of providing convincing evidence of clinical or scientific impact due to its very small, homogeneous sample, limited validation strategy, and overstatement of results. I suggest to consider to expand the paper with a validation on a larger and more diverse cohort, including clinical populations. a stronger and more rigorous gold standard for distance and stride length and a more cautious interpretation of statistical results and limitations.
Author Response
For research article
|
Response to Reviewer 2 Comments
|
||
|
1. Summary |
|
|
|
The authors thank you for taking the time to review this manuscript. Your comments have helped us improve our manuscript. We have addressed each of your comments in our responses.
|
||
|
2. Questions for General Evaluation |
Reviewer’s Evaluation |
Response and Revisions Thank you for your constructive comments |
|
Does the introduction provide sufficient background and include all relevant references? |
Yes/Can be improved/Must be improved/Not applicable |
|
|
Are all the cited references relevant to the research? |
Yes/Can be improved/Must be improved/Not applicable |
|
|
Is the research design appropriate? |
Yes/Can be improved/Must be improved/Not applicable |
|
|
Are the methods adequately described? |
Yes/Can be improved/Must be improved/Not applicable |
|
|
Are the results clearly presented? |
Yes/Can be improved/Must be improved/Not applicable |
|
|
Are the conclusions supported by the results? |
Yes/Can be improved/Must be improved/Not applicable |
|
|
3. Point-by-point response to Comments and Suggestions for Authors |
||
|
Comments 1: The study addresses the estimation of stride length and total distance covered through wearable sensors. Accurate, unobtrusive gait assessment tools are highly valuable in clinical rehabilitation, sports science, and public health. The paper introduces two new model-based approaches (director coefficients of acceleration data and Dynamic Time Warping), expanding beyond conventional accelerometer-only solutions. The manuscript is generally well structured and easy to read. However, there are several major concerns. Some of these are acknowledged by the authors as limitations, but they represent serious methodological weaknesses. Major concerns
|
||
|
Response 1: The small sample size of this study does not enable the results to be generalized, although similar sample sizes are reported in the literature [69, 74-77], see lines 389 to 391 of the manuscript. Before being used for real-world clinical practice, the approaches of this study will be assessed in people with gait impairments (stroke, Parkinson’s disease, frailty, etc.). The assessment of these approaches in healthy participants will be followed by assessments in participants with impairments. See lines 396 to 398. |
||
|
Comments 2: The reference (“multiplying round trips × 40 m”) ignores turning maneuvers and the geometry of the walking path. This introduces a systematic bias that is openly acknowledged but not properly corrected. Response 2: We completely agree with your comment. To correct this systematic bias, it would have been necessary to use, for example, an odometer to track the participant in order to determine the actual total distance covered. However, the odometer might have prevented the participant from walking at their maximum speed. The ideal solution would have been to use a motion capture system, which we recommend for further studies, see line 392 to 395.
|
||
|
Comments 3: The model is derived from a single participant’s motion capture data, which raises questions about representativeness and robustness. |
||
|
Response 3: We agree with your opinion. As mentioned in the limitations section (discussion), we are aware that the model based on a single participant’s motion capture date is neither representative nor robust as mentioned on lines 382 to 384.
Comments 4: On what basis is the initial hypothesis made that the DTW methodology would be more accurate than the approach based on director coefficients of acceleration data? This requires proper justification.
Response 4: DTW is an algorithm that measures the similarity between two temporal sequences by finding an optimal alignment that minimizes the cumulative distance between corresponding points regardless of speed and duration, whereas the acceleration director co-efficient is a mathematical model (see page 3, lines 110 to 114).
Comments 5: A clearer justification of the novelty and added value of the proposed algorithms is needed.
Response 5: Firstly, the novel of this study is identification of toe-off and Heel-Strike events using both acceleration and pressure signals. Next, compared to the approaches reported in the literature, the novel of proposed algorithms is use of DTW and director coefficient of antero-posterior acceleration to estimate stride length.
Comments 6: Statistical power is insufficient, and the conclusions about “excellent” validity are overstated. The reliance on ICCs with very narrow confidence intervals seems overly optimistic given the small sample size. Non-parametric tests (Mann-Whitney) are used, but the choice is not well justified, nor is there any adjustment for multiple comparisons.
Response 6: The information was added on lines 251 to 255 and 411 to 413 (conclusion). “Since the sample size is small and the data distribution is non-normal, non-parametric statistical analyses were performed, in particular the Mann-Whitney U tests. the Mann-Whitney U tests were used to determine whether there is a statistically significant difference between 1) Approach-1 and Approach-2; 2) actual total distance and estimated distance from smart insole and Gait Up”, …” Although the sample size was small and the reference system was no robust, the results showed that the accuracies and ICC values were very high for stride length and total distance estimations using both approaches” see in statistical analysis and conclusion sections
Comments 7: The discussion is very limited and should be expanded to cover at least (but not only) clinical aspects and the interpretation of results. While technical accuracy is reported, clinical utility is not demonstrated. Would these methods provide meaningful improvements in patient monitoring compared to existing commercial solutions (e.g., IMU-based gait monitors)? The study does not define a minimal clinically important difference (MCID) for distance estimation, making it unclear whether a 1–2% gain in accuracy has any practical relevance.
Response 7: The discussion was expanded, see Pages 14 to 16. Given that the study participants were healthy people and that the aim of this study was to develop and assess the accuracy of novel approaches, we believe it would be premature to assess the minimal clinically important difference (MCID) in this study.
Comments 8: Minor concerns
Response 8: The correction has been made see line 58.
Comments 9: L114: Please include information about the producer the first time the sensors are mentioned. Response 9: The correction has been made see line 120
Comments 10: L116: Walking speed range is a result; it should be included in the Results section, not in Materials and Methods.
Response 10: The correction has been made see line 117 to 119. Comments 11: L120: The “gold standard” for total distance (multiplying round trips × 40 m) ignores turning maneuvers and the geometry of the walking path. This introduces a systematic bias that is acknowledged but not properly corrected.
Response 11: Recommendations were made in the discussion section see 392 to 394. Comments 12: L130: Please provide more details when describing the smart insoles. Report technical characteristics of the embedded sensors and the insole material or provide a reference paper. Are the insoles self-made or commercial?
Response 12: A photo of smart insole has been added in the manuscript, see figure 2 , line 151. The smart insole used in this manuscript is commercially available.
Figure 2. Sites of sensors under of the smart insoles
Comments 13: You mention horizontal acceleration. This is not a good description; I assume you mean antero-posterior acceleration. In the figures, acceleration along the x-axis is reported, but the reference system is never defined. I suggest including a reference system in Figure 1 or 3 and replacing the term horizontal acceleration with antero-posterior acceleration. Response 13: The correction has been made see line 160 Comments 14: L154–160: It is not useful to list the specific Matlab functions. A description of the implemented algorithm is sufficient.
Response 14: Some reviewers recommended including specific Matlab functions in the manuscript. Comments 15: Figures 4 and 5: Gait events are indicated, but it is unclear where toe-offs or heel-strikes occur. Please mark them directly on the graph. Measurement units for the pressure signal are missing. The caption of Figure 4 should read identification rather than segmentation. In Figure 5, you claim to show a gait cycle, but the trend does not appear to represent a complete cycle. In particular, the initial and final pressure values are very different.
Response 15: Figures 4 , 5 and 6 were improved see lines 179, 182 and 194 Comments 16: L167–168: “Six markers were positioned on the front and back feet of a participant.” This is too generic; please specify the anatomical landmarks where the markers were placed.
Response 16: The correction has been made 186 Comments 17: Figure 6: Same issues as Figures 4 and 5—unclear whether this represents a complete cycle, missing measurement units, unclear identification of heel-strike and toe-off. The graphs represent marker positions, but which marker? And according to which reference system (not defined)? Response 17: The correction has been made see lines 179, 182 and 194 Comments 18: L240–242: The number of participants, demographic, and anthropometric data should be reported in Materials and Methods, including sex/gender. Response 18: The correction has been made Comments 19: Table 1: Units of measurement are missing. There is also confusion between stride and step. Response 19: The correction has been made see lines 117 to 119 Response to Comments on the Quality of English Language |
||
|
Point 1: |
||
|
Response 1: The quality of English Language was improved in the manuscript in red. |
||

Reviewer 3 Report
Comments and Suggestions for Authors
The authors of the manuscript proposed for review use a smart insole with pressure (force) sensors and an accelerometer to validate two approaches created by them, based on the step length model, to estimate the distance travelled by the number of steps. Experiments were conducted and the results were processed according to the proposed algorithms. The authors claim that the results did not give a statistically significant difference between the actual distance and those calculated using the smart insole.
The article is properly structured, written in an acceptable scientific style and contains the required number of literary references.
I have the following comments on the article written in this way:
- The introduction states (lines 83-88) that accelerometer measurements lead to an accumulation of errors due to drift, discretization and some features of the accelerometer's operating principle itself. The double integration of these errors leads to unacceptable deviations. What is new in the two proposed methods that eliminate these errors? In general, could the authors indicate in the introduction what new features they offer with their methods? Which shortcomings of the previous methods have been overcome?
- Measurements using the smart insole depend on the type of sensors used. I ask the authors to specify the type of accelerometer and pressure sensors they used. It is necessary to know what type of accelerometer and pressure sensors are used, whether they are capacitive, piezoelectric, piezoresistive, etc. Also, some of their main characteristics, such as sensitivity, frequency band, measurement range, should be specified.
- In addition to the "Sites of sensors under the smart insoles" shown in Figure 2, it is desirable to show a detailed photo of the smart insole itself to get a clearer idea of ​​its overall appearance.
- The sensors shown in Figure 2 are connected to a power supply and a transmitter. Can you specify the connections and the location on the insole of these elements?
- In approach 1 of Stride length model-based algorithms, you practically assume that the horizontal acceleration of the accelerometer can be approximated by a linear function. What gives you reason for this assumption? Aren't you assuming a large error from entering the input data in this way? Please estimate this error, for example using the horizontal acceleration in Figure 5.
- In my opinion "Approach-2" is not described clearly enough. Please, although you cited [63], please explain the operator DTW(Amxn(t),Asxn(t)) in detail, as well as the meaning and type of all symbols shown in this method. I mean the symbols in formulas (6) to (8).
- During normal walking, and even more so when patients with health problems walk, the soles of their feet can perform complex spatial movements. This results in data appearing along the three axes of the accelerometer. In addition, there will be accompanying rotations. How do your "Stride length model-based algorithms" take these movement features into account?
- In my opinion, the conclusion is incomplete and needs to be revised. I would ask the authors to justify their contributions more clearly, and to present accurate quantitative data in support of their arguments.
Author Response
For research article
|
Response to Reviewer 3 Comments
|
||
|
1. Summary |
|
|
|
The authors thank you for taking the time to review this manuscript. Your comments have helped us improve our manuscript. We have addressed each of your comments in our responses.
|
||
|
2. Questions for General Evaluation |
Reviewer’s Evaluation |
Response and Revisions |
|
Does the introduction provide sufficient background and include all relevant references? |
Yes/Can be improved/Must be improved/Not applicable |
[Please give your response if necessary. Or you can also give your corresponding response in the point-by-point response letter. The same as below] |
|
Are all the cited references relevant to the research? |
Yes/Can be improved/Must be improved/Not applicable |
|
|
Is the research design appropriate? |
Yes/Can be improved/Must be improved/Not applicable |
|
|
Are the methods adequately described? |
Yes/Can be improved/Must be improved/Not applicable |
|
|
Are the results clearly presented? |
Yes/Can be improved/Must be improved/Not applicable |
|
|
Are the conclusions supported by the results? |
Yes/Can be improved/Must be improved/Not applicable |
|
|
3. Point-by-point response to Comments and Suggestions for Authors |
||
|
Comments 1: The authors of the manuscript proposed for review use a smart insole with pressure (force) sensors and an accelerometer to validate two approaches created by them, based on the step length model, to estimate the distance travelled by the number of steps. Experiments were conducted and the results were processed according to the proposed algorithms. The authors claim that the results did not give a statistically significant difference between the actual distance and those calculated using the smart insole. The introduction states (lines 83-88) that accelerometer measurements lead to an accumulation of errors due to drift, discretization and some features of the accelerometer's operating principle itself. The double integration of these errors leads to unacceptable deviations. What is new in the two proposed methods that eliminate these errors? In general, could the authors indicate in the introduction what new features they offer with their methods? Which shortcomings of the previous methods have been overcome?
|
||
|
Response 1: Thank you for your constructive comments. In this study, we did not drift acceleration to avoid accumulating errors. Instead, we used the polynomial signal. The special feature of our study is the use of the dynamic time warping (DTW) algorithm that measures the similarity between two temporal sequences by finding an optimal alignment that minimizes the cumulative distance between corresponding points regardless of speed and duration. We added in the manuscript on line 110 to 114.
|
||
|
Comments 2: Measurements using the smart insole depend on the type of sensors used. I ask the authors to specify the type of accelerometer and pressure sensors they used. It is necessary to know what type of accelerometer and pressure sensors are used, whether they are capacitive, piezoelectric, piezoresistive, etc. Also, some of their main characteristics, such as sensitivity, frequency band, measurement range, should be specified.
|
||
|
Response 2: The smart insole used in this study is commercial insole. Unfortunately, we do not have this information. Comments 3: In addition to the "Sites of sensors under the smart insoles" shown in Figure 2, it is desirable to show a detailed photo of the smart insole itself to get a clearer idea of ​​its overall appearance.
Response 3: A photo of the smart insole has been added in the manuscript on line 151.
Figure 2. Sites of sensors under of the smart insoles
Comments 4: The sensors shown in Figure 2 are connected to a power supply and a transmitter. Can you specify the connections and the location on the insole of these elements?
Response 4: “The smart insole is a small, low-power, stand-alone device that integrates a microcontroller, Bluetooth communication, and battery. The electronic module is in the midfoot.” The information was added on lines 147 and 148.
Comments 5: In approach 1 of Stride length model-based algorithms, you practically assume that the horizontal acceleration of the accelerometer can be approximated by a linear function. What gives you reason for this assumption? Aren't you assuming a large error from entering the input data in this way? Please estimate this error, for example using the horizontal acceleration in Figure 5.
Response 5: We assume that the antero-posterio acceleration can be approximated by a linear function based on studies of Zrenner et al. 2018 and Gradl et al. 2018. In our manuscript, there is mentioned that: “The calculation of the foot velocity used in Zrenner et al. [10] was based on a polynomial function of second-order, where the constants were estimated on a set of training data with known reference velocity observations using parametric regression analysis [45] » See lines 95 to 98.
Comments 6: In my opinion "Approach-2" is not described clearly enough. Please, although you cited [63], please explain the operator DTW(Amxn(t),Asxn(t)) in detail, as well as the meaning and type of all symbols shown in this method. I mean the symbols in formulas (6) to (8).
Response 6: The correction has been made in the manuscript line 233 to 240. “the coefficient (S) for n samples in (6) is calculated as follows: S = 1/||Asxn|| (6)
Where ||Asxn|| represents the absolute value of the antero-posterior acceleration during the swing phase. The estimation of the DTW value (d(t)) between the acceleration data of the reference model (Amxn(t)) and the smart insole (Asxn(t)) for n samples is il-lustrated below: dn(t) = DTW(Amxn(t),Asxn(t)) (7) E = mean(dn(t))S (8) Where Amxn represents the antero-posterior acceleration of the reference model for n samples, Asxn represents the antero-posterior acceleration of the smart insole for n samples. The average stride length (L) is calculated by multiplying E by the position Px of the model as illustrated in the Algorithm 2.”
Comments 7: During normal walking, and even more so when patients with health problems walk, the soles of their feet can perform complex spatial movements. This results in data appearing along the three axes of the accelerometer. In addition, there will be accompanying rotations. How do your "Stride length model-based algorithms" take these movement features into account?
Response 7: We agree with your opinion. As indicated in the limitation section on lines 392 to 394, to better characterize the movement, it was necessary to use both the gyroscope to consider rotational movements and the three axes of acceleration. This is a limitation of the study. We made recommendations in the discussion section to consider the gyroscope in the further study.
Comments 8: In my opinion, the conclusion is incomplete and needs to be revised. I would ask the authors to justify their contributions more clearly, and to present accurate quantitative data in support of their arguments.
Response 8: The correction has been made in the manuscript on lines 411 to 417:“This study enabled the development and validation of two novel model-based approaches for measuring stride length for total distance estimation using a smart insole integrating accelerometer and pressure sensors in healthy participants. These two approaches are based on the director coefficients of the acceleration data (Approach-1) and dynamic time warping (Approach-2). Although the sample size was small and the reference system was no robust, the results showed that the accuracies and ICC values were very high for stride length and total distance estimations using both approaches. These approaches yielded similar results. Compared to model-based approaches in the literature, these two novel approaches appear to be more accurate for stride length and total distance estimation. However, further studies should be conducted, considering the limitations of this study.”
|
||
|
4. Response to Comments on the Quality of English Language |
||
|
Point 1: |
||
|
Response 1: The quality of English Language was improved in the manuscript in red. |
||

Round 2
Reviewer 1 Report
Comments and Suggestions for Authors
The authors have addressed my previous comments and concerns sufficiently, and I appreciate their detailed responses.
Reviewer 3 Report
Comments and Suggestions for Authors
The authors have taken all the comments into account. The changes made to the manuscript have improved the quality and fully satisfy my critical comments. Therefore, I propose that the modified version of the article be published,